# Nitropyridines as 2π-Partners in 1,3-Dipolar Cycloadditions with N-Methyl Azomethine Ylide: An Easy Access to Condensed Pyrrolines

**DOI:** 10.3390/molecules26185547

**Published:** 2021-09-13

**Authors:** Maxim A. Bastrakov, Alexey K. Fedorenko, Alexey M. Starosotnikov, Alexander Kh. Shakhnes

**Affiliations:** N.D. Zelinsky Institute of Organic Chemistry RAS, Leninsky Prosp. 47, 119991 Moscow, Russia; alexeyfedorenko21@mail.ru (A.K.F.); alexey41@list.ru (A.M.S.); shakhnes@ioc.ac.ru (A.K.S.)

**Keywords:** nitro group, nitropyridines, pyrrolo[3,4-*c*]pyridines, dearomatization, 1,3-dipolar cycloadditition, pyrrolidines, azomethine ylides

## Abstract

1,3-Dipolar cycloaddition reactions of 2-substituted 5-*R*-3-nitropyridines and isomeric 3-*R*-5-nitropyridines with N-methyl azomethine ylide were studied. The effect of the substituent at positions 2 and 5 of the pyridine ring on the possibility of the [3+2]-cycloaddition process was revealed. A number of new derivatives of pyrroline and pyrrolidine condensed with a pyridine ring were synthesized.

## 1. Introduction

Pyridine derivatives of both natural and synthetic origin are one of the promising classes of heterocyclic compounds, which have a great synthetic potential and are of interest for the synthesis of biologically active substances. It is known that a large number of pyridine derivatives possess antimicrobial, antiviral, anticancer, analgesic, and antioxidant activities [1,2,3,4,5]. Recently, there has been a rapid development of approaches to the synthesis (including industrial) of condensed pyridines with various types of useful biological activity [6,7,8,9,10,11,12].

1,3-Dipolar cycloaddition reactions represent one of the most effective strategies leading to new fused heterocycles. Cycloaddition to aromatic compounds is accompanied by dearomatization, which leads to saturated, hardly accessible structures [13,14,15,16,17,18].

Earlier, we reported on the development of a method for the synthesis of pyrrolidine and pyrroline derivatives condensed with a pyridine ring on the basis of 2-*R*-3,5-dinitropyridines [19,20]. It was shown that, depending on the nature of 2-R, annulation of either two pyrrolidine rings or one pyrroline ring is possible [20], Figure 1.

## 2. Results and Discussion

### 2.1. Synthesis of Starting 2-Substitutied 5-R-3nitropyridines ***2a–q***

In this work we studied the effect of the substituent at position 5 of the pyridine ring on the [3+2]-cycloaddition process. We used available 2-chloro-5-*R*-3-nitropyridines **1** as starting compounds. The mobile chlorine atom in these compounds is capable of being replaced by the action of nucleophiles under mild conditions. As a result of reaction **1** with thiols, amines, and phenols, compounds **2a–q** were synthesized (Figure 2, Table 1).

### 2.2. [3+2]-Cycloaddition of Nitropyridines ***2***

We studied pyridines **2a–q** in 1,3-dipolar cycloaddition reactions with an excess of N-methyl azomethine ylide **3**, which was generated in situ from sarcosine and paraformaldehyde under reflux in toluene. In the case of pyridines **2a,b,d,e,g–i,k–m,o–q**, a dipole was added at the C=C-NO_2_ bond followed by elimination of the HNO_2_ molecule and rearomatization of the system with the formation of pyrroline derivatives **4**, Figure 3.

Thus, pyridines containing a donor substituent (Me or H) at position 5 do not undergo a [3+2]-cycloaddition reaction. In addition, in the presence of an amine moiety in position 2 (compounds **2c, 2j, 2n, 2q**), the reaction also does not proceed. This reactivity is most likely associated with the electron-donor effect of the amino group, which reduces the overall electrophilicity of the starting compound [20,21]. The target product was isolated only for compound **2f** where conversion of the starting pyridine was about 50% according to the NMR spectroscopy data.

The reaction of pyridine **2o** with azomethine ylide **3** deserves special attention (Figure 4). We found that this reaction resulted in the addition of two molecules of a dipole to the pyridine nucleus with the formation of compound **5**.

It is known that, in similar reactions, the addition of a dipole occurs from the opposite sides of the benzene (pyridine) ring, providing *trans*-cycloadducts [19,20,22,23]. However there are examples of the formation of *cis*-cycloadducts [24]. The cycloaddition of **3** to pyridine **2o** can in principle provide two isomeric products **5** and **6,**
Figure 1. We carried out a detailed study of the cycloaddition adduct using various NMR experiments (COSY, ^1^H-^13^C HMBC, ^1^H-^13^C HSQC, NOESY). The full assignment of hydrogen and carbon atoms in NMR spectra was made. The ^1^H-^1^H NOESY spectrum of **5** contains a characteristic cross-peak corresponding to the interaction of spatially close H(1) and H(6) protons, Figure 1. At the same time, the interaction between H(1) and H(5) was not observed. These data allowed us to unambiguously confirm trans-cycloaddition and the formation of compound **5.**

## 3. Materials and Methods

### 3.1. General Information

All chemicals were of commercial grade and used directly without purification. Melting points were measured on a Stuart SMP 20 apparatus (Stuart (Bibby Scientific), UK). ^1^H and ^13^C NMR spectra were recorded on a Bruker AM-300 (at 300.13 and 75.13 MHz, respectively, Bruker Biospin, Germany) and a Bruker Avance DRX 500 (at 500 and 125 MHz, respectively, Bruker Biospin, Germany) in DMSO-*d*_6_ or CDCl_3_. HRMS spectra were recorded on a Bruker micrOTOF II mass spectrometer (Bruker micrOTOF II mass spectrometer) using ESI. All reactions were monitored by TLC analysis using ALUGRAM SIL G/UV254 plates, which were visualized by UV light (see Appendix A).

### 3.2. Synthesis of Compounds ***2a*, *2d*, *2g*, *2h*, *2k*, *2l*, *2o***

An appropriate thiol (1 mmol) and Et_3_N (0.14 mL, 1 mmol) were added to a solution of 1 mmol of appropriate chloride in methanol (30 mL). The reaction mixture was stirred at room temperature for 0.5–2 h until the starting compound was completely consumed (TLC). Then mixture was poured into water and acidified with HCl to a pH of 4. The precipitate that formed was filtered off, washed with water, and dried in air.

*2-(Benzylthio)-3-nitro-5-(trifluoromethyl)pyridine* (**2a**). 88% M.p. 58–60 °C. ^1^H NMR (300 MHz, CDCl_3_): δ 4.54 (s, 2H, CH_2_), 7.30–7.46 (m, 5H, Ph), 8.73 (s, 1H_._), 8.97 (s, 1H). NMR (75 MHz, CDCl_3_): δ 35.8, 120.8, 122.1, 122.5, 124.4, 127.6, 128.5, 128.7, 129.4, 131.1, 131.1, 136.0, 140.8, 149.2, 149.3, 162.1. HRMS (ESI) calc. for [C_13_H_9_F_3_N_2_O_2_S]^+^ [M + H]^+^ 315.0418, found 315.0410.

*Methyl 6-(benzylthio)-5-nitronicotinate* (**2d**). 95%. M.p. 103–105 °C. ^1^H NMR (300 MHz, CDCl_3_): δ 4.00 (s, 3H, CH_3_), 4.53 (s 2H, CH_2_), 7.26–7.44 (m, 5H, Ph), 8.01 (s, 1H_._), 8.25 (s, 1H). ^13^C NMR (75 MHz, CDCl_3_): δ 35.8, 52.9, 121.8, 127.5, 128.6, 129.4, 134.4, 136.2, 141.1, 153.2, 162.3, 164.0. HRMS (ESI) calc. for [C_14_H_12_N_2_O_4_S]^+^ [M + H]^+^ 305.0590, found 305.0591.

*Methyl 6-(isobutylthio)-5-nitronicotinate* (**2g**). 85%. M.p. 64–66 °C. ^1^H NMR (300 MHz, CDCl_3_): δ 1.08 (d, *J* = 6.6 Hz, 6H, 2CH_3_(i-Bu)), 1.94–2.04 (m, 1H, CH (i-Bu)), 3.19 (d, *J* = 6,6 Hz, 2H, CH_2_ (i-Bu)), 3.99 (s, 2H, CO_2_CH_3_), 8.98 (s, 1H, C(4)-H), 9.20 (s, 1H, C(6)-H). ^13^C NMR (75 MHz, CDCl_3_): δ 22.2, 27.9, 39.4, 52.8, 121.2, 134.3, 141.6, 153.1, 163.2, 164.1. HRMS (ESI) calc. for [C_11_H_14_N_2_O_4_S]^+^ [M + H]^+^ 271.0751, found 271.0747.

*2-(Benzylthio)-5-chloro-3-nitropyridine* (**2h**). 77%. M.p. 65–67 °C. ^1^H NMR (300 MHz, CDCl_3_): δ 4.48 (s, 2H, CH_2_), 7.26–7.45 (m, 5H, Ph), 8.51 (d, *J* = 2.4 Hz, 1H, C(4)-H), 8.70 (d, *J* = 2.4 Hz, 1H, C(6)-H). ^13^C NMR (75 MHz, CDCl_3_): δ 35.5, 127.0, 127.5, 128.6, 129.3, 133.1, 136.4, 141.3, 151.9, 155.9. HRMS (ESI) calc. for [C_12_H_9_ClN_2_O_2_S]^+^ [M + H]^+^ 281.0143, found 281.0146.

*2-(Benzylthio)-5-methyl-3-nitropyridine* (**2k**). 58%. M.p. 83–85 °C. ^1^H NMR (300 MHz, CDCl_3_): δ 2.43 (s, 3H, CH_3_), 4.48 (s, 2H, CH_2_), 7.25–7.45 (m, 5H, Ph), 8.32 (s, 1H), 8.58(s, 1H). ^13^C NMR (75 MHz, CDCl_3_): δ 17.3, 35.2, 127.2, 127.4, 128.5, 129.2, 129.4, 133.8, 137.1, 141.4, 153.8. HRMS (ESI) calc. for [C_13_H_12_N_2_O_2_S]^+^ [M + H]^+^ 261.0692, found 261.0687.

*2-(Benzylthio)-5-bromo-3-nitropyridine* (**2l**). 68%. M.p. 33–35 °C. ^1^H NMR (300 MHz, CDCl_3_): δ 4.38 (s, 2H, CH_2_), 7.18–7.37 (m, 5H, Ph), 8.56 (d, *J* = 2.1 Hz, 1H, C(4)-H), 8.70 (d, *J* = 2.1 Hz, 1H, C(6)-H). ^13^C NMR (75 MHz, CDCl_3_): δ 35.5, 114.4, 127.5, 128.6, 129.4, 135.8, 136.4, 142.0, 154.0, 156.4. HRMS (ESI) calc.for [C_12_H_9_BrN_2_O_2_S]^+^ [M + H]^+^ 324.9639, found 324.9641.

*Methyl 2-(benzylthio)-5-nitronicotinate* (**2o**). 98%. M.p. 103–105 °C. ^1^H NMR (300 MHz, CDCl_3_): δ 3.91 (s, 3H, CH_3_), 4.44 (s, 2H, CH_2_), 7.16–7.37 (m, 5H, Ph), 8.88 (s, 1H), 8.30(s, 1H). ^13^C NMR (75 MHz, CDCl_3_): δ 35.8, 53.0, 122.1, 127.4, 128.6, 129.4, 133.5, 136.5, 140.2, 146.5, 163.9, 169.6. HRMS (ESI) calc. for [C_14_H_12_N_2_O_4_S]^+^ [M + H]^+^ 305.0592, found 305.0591.

### 3.3. Synthesis of Compounds ***2b*, *2e*, *2i*, *2m*, *2p***

4-Nitrophenol 0.14g (1 mmol) and Na_2_CO_3_ (0.106 g, 1 mmol) were added to a solution of appropriate chloride (1 mmol) in acetonitrile (15 mL). The reaction mixture was refluxed for 2 h until the starting compound was completely consumed (monitoring by TLC), poured into a five-fold excess of water, and acidified with HCl to a pH of 2. The precipitate that formed was filtered off, washed with water, and dried in air.

*3-Nitro-2-(4-nitrophenoxy)-5-(trifluoromethyl)pyridine* (**2b**). 79%. M.p. 96–98 °C. ^1^H NMR (300 MHz, CDCl_3_): δ 7.41 (d, *J* = 8.7 Hz, 2H, 4-NO_2_-Ph), 8.39 (d, *J* = 8.7 Hz, 2H, 4-NO_2_-Ph), 8.62 (s, 1H, C(4)-H), 8.69 (s, 1H, C(6)-H). ^13^C NMR (75 MHz, CDCl_3_): δ 120.3, 122.6, 122.9, 123.4, 123.9, 125.6, 133.7, 145.8, 148.8, 156.6. HRMS (ESI) calc. for [C_12_H_6_F_3_N_3_O_5_]^+^ [M + H]^+^ 330.0330, found 330.0332.

*Methyl 5-nitro-6-(4-nitrophenoxy)nicotinate* (**2e**). 93%. M.p. 128–130 °C. ^1^H NMR (300 MHz, CDCl_3_): δ 4.00 (s, 3H, CO_2_CH_3_), 7.39 (d, *J* = 9.0 Hz, 2H, 4-NO_2_-Ph), 8.36 (d, *J* = 9.0 Hz, 2H, 4-NO_2_-Ph), 8.93 (s, 1H, C(4)-H), 8.97 (s, 1H, C(6)-H). ^13^C NMR (75 MHz, CDCl_3_): δ 53.0, 122.6, 122.8, 125.9, 134.0, 136.9, 145.6, 152.9, 156.8, 163.2. HRMS (ESI) calc. for [C_13_H_9_N_3_O_7_]^+^ [M + H]^+^ 320.0503, found 320.0513.

*5-Chloro-3-nitro-2-(4-nitrophenoxy)pyridine* (**2i**). 81%. M.p. 110–112 °C. ^1^H NMR (300 MHz, CDCl_3_): δ 7.37 (d, *J* = 9.0 Hz, 2H, 4-NO_2_-Ph), 8.34–8.37 (m, 3H, C(4)-H и 4-NO_2_-Ph), 8.44 (d, *J* = 2.1 Hz, 1H, C(6)-H). ^13^C ЯMP (75 MHz, CDCl_3_): δ 122.2, 125.6, 126.9, 134.3, 135.4, 145.4, 150.2, 153.0, 157.2. HRMS (ESI) calc. for [C_11_H_6_ClN_3_O_5_]^+^ [M + H]^+^ 317.9882, found 317.9888.

*5-Bromo-3-nitro-2-(4-nitrophenoxy)pyridine* (**2m**). 74%. M.p. 118–120 °C. ^1^H NMR (300 MHz, CDCl_3_): δ 7.29 (d, *J* = 9.0 Hz, 2H, 4-NO_2_-Ph), 8.27 (d, *J* = 9.0 Hz, 2H, 4-NO_2_-Ph), 8.48 (d, *J* = 2.1 Hz, 1H, C(4)-H), 8.63 (d, *J* = 2.1 Hz, 1H, C(6)-H). ^13^C NMR (75 MHz, CDCl_3_): δ 113.8, 122.2, 125.6, 138.1, 145.4, 152.4, 153.5, 157.2. HRMS (ESI) calc. for [C_11_H_6_BrN_3_O_5_]^+^ [M + H]^+^ 361.9379, found 361.9383.

*Methyl 5-nitro-2-(4-nitrophenoxy)nicotinate* (**2p**). 92%. M.p. 148–150 °C. ^1^H NMR (300 MHz, CDCl_3_): δ 3.96 (s, 3H, CO_2_CH_3_), 7.29 (d, *J* = 8.7 Hz, 2H, 4-NO_2_-Ph), 8.28 (d, *J* = 8.7 Hz, 2H, 4-NO_2_-Ph), 9.01 (s, 2H, C(4)-H и C(6)-H). ^13^C NMR (75 MHz, CDCl_3_): δ 53.3, 115.3, 122.6, 125.6, 137.6, 140.5, 145.5, 147.0, 157.3, 162.6, 163.3. HRMS (ESI) calc for [C_13_H_9_N_3_O_7_]^+^ [M + H]^+^ 320.0505, found 320.0513.

### 3.4. Synthesis of Compounds ***2c*, *2f*, *2j*, *2n*, *2q***

Pyrrolidine 0.16 mL (2 mmol) was added to a solution of appropriate chloride (1 mmol) in methanol (15 mL). The reaction mixture was stirred at room temperature for 2 h (monitoring by TLC), poured into an excess of water, and acidified with HCl to a pH of 2. The precipitate that formed was filtered off, washed with water, and dried.

*3-Nitro-2-(pyrrolidin-1-yl)-5-(trifluoromethyl)pyridine* (**2c**) 95%. M.p. 65–67 °C. ^1^H NMR (300 MHz, CDCl_3_): δ 2.04 (t, *J* = 6.6 Hz, 4H, CH_2_CH_2_CH_2_CH_2_), 3.47 (br.s, 4H, CH_2_CH_2_CH_2_CH_2_), 8.31 (s, 1H, C(4)-H), 8.55 (s, 1H, C(6)-H). ^13^C NMR (75 MHz, CDCl_3_): δ 25.3, 49.8, 113.5, 114.0, 121.7, 125.4, 130.1, 132.4, 132.5, 148.5, 148.6, 151.0. HRMS (ESI) calc. for [C_10_H_10_F_3_N_3_O_2_]^+^ [M + H]^+^ 262.0797, found 262.0798.

*Methyl 5-nitro-6-(pyrrolidin-1-yl)nicotinate* (**2f**). 92%. M.p. 87–89 °C. ^1^H NMR (300 MHz, CDCl_3_): δ 2.01 (br.s, 4H, CH_2_CH_2_CH_2_CH_2_), 3.48 (br.s, 4H, CH_2_CH_2_CH_2_CH_2_), 3.91 (s, 3H, CO_2_CH_3_), 8.62 (s, 1H, C(4)-H), 8.88 (s, 1H, C(6)-H). ^13^C NMR (75 MHz, CDCl_3_): δ 25.3, 49.8, 52.1, 113.6, 130.7, 136.1, 151.3, 153.1, 164.9. HRMS (ESI) calc. for [C_11_H_13_N_3_O_4_]^+^ [M + H]^+^ 252.0978, found 252.0689.

*5-Chloro-3-nitro-2-(pyrrolidin-1-yl)pyridine* (**2j**). 91%. M.p. 73–75 °C. ^1^H NMR (300 MHz, CDCl_3_): δ 1.98–2.05 (m, 4H, CH_2_CH_2_CH_2_CH_2_), 3.40 (t, *J* = 6.6 Hz, 4H, CH_2_CH_2_CH_2_CH_2_), 8.09 (d, *J* = 2.4 Hz, 1H, C(4)-H), 8.29 (d, *J* = 2.4 Hz, 1H, C(6)-H). ^13^C NMR (75 MHz, CDCl_3_): δ 25.4, 49.7, 117.0, 130.8, 133.8, 148.8, 150.5. HRMS (ESI) calc. for [C_9_H_10_ClN_3_O_2_]^+^ [M + H]^+^ 228.0534, found 228.0534.

*5-Bromo-3-nitro-2-(pyrrolidin-1-yl)pyridine* (**2n**): 86%. M.p. 67–69 °C. ^1^H NMR (300 MHz, CDCl_3_): δ 1.92 (dd, *J*_1_ = 12.3 Hz, *J*_2_ = 6.6 Hz, 4H, CH_2_CH_2_CH_2_CH_2_), 3.31 (t, *J* = 6.6 Hz, 4H, CH_2_CH_2_CH_2_CH_2_), 8.13 (d, *J* = 2.1 Hz, 1H, C(4)-H), 8.27 (d, *J* = 2.1 Hz, 1H, C(6)-H). ^13^C ЯMP (75 MHz, CDCl_3_): δ 25.4, 49.7, 103.5, 136.4, 149.0, 152.5. HRMS (ESI) calc. for [C_9_H_10_BrN_3_O_2_]^+^ [M + H]^+^ 272.0029, found 272.0026.

*Methyl 5-nitro-2-(pyrrolidin-1-yl)nicotinate* (**2q**): 79%. M.p. 96–98 °C. ^1^H NMR (300 MHz, CDCl_3_): δ 1.93 (br.s, 4H, CH_2_CH_2_CH_2_CH_2_), 3.45 (br.s, 4H, CH_2_CH_2_CH_2_CH_2_), 3.85 (s, 3H, CO_2_CH_3_), 8.57 (d, *J* = 2.1 Hz, 1H, C(4)-H), 9.00 (d, *J* = 2.1 Hz, C(6)-H). ^13^C ЯMP (75 MHz, CDCl_3_): δ 25.4, 50.4, 52.7, 108.9, 133.8, 135.5, 147.5, 156.6, 165.9. HRMS (ESI) calc. for [C_12_H_6_F_3_N_3_O_5_]^+^ [M + H]^+^ 330.0330, found 330.0332.

### 3.5. Synthesis of Compounds ***4a–j*, *5***

A mixture of an appropriate nitropyridine (1 mmol), paraformaldehyde (0.18 g, 6 mmol), and sarcosine (0.50 g, 6 mmol) was refluxed in toluene (30 mL), until the starting compound was completely consumed (TLC), and the mixture was cooled to room temperature and filtered. The solvent was evaporated under reduced pressure, and the residue was washed with hexane. The precipitate was filtered off. For compound **5**, the residue was purified by column chromatography on MN Kieselgel 60 (0.04–0.063 mm/230–400 mesh) using EtOAc:MeOH (10:1) as an eluent.

*4-(Benzylthio)-2-methyl-7-(trifluoromethyl)-2,3-dihydro-1H-pyrrolo[3,4-c]pyridine* (**4a**). 59%. M.p. 56–58 °C. ^1^H NMR (300 MHz, CDCl_3_): δ 2.61 (s, 3H, N-CH_3_), 3.85 (s, 2H, N-CH_2_), 4.08 (s, 2H, N-CH_2_), 4.56 (s, 2H, S-CH_2_), 7.24–7.45 (m, 5H, Ph), 8.57 (s, 1H, C(6)-H). ^13^C NMR (75 MHz, CDCl_3_): δ 33.6, 42.0, 58.1, 59.7, 117.0, 117.4, 117.8, 118.3, 118.6, 122.2, 125.9, 127.3, 128.6, 129.1, 129.5, 134.3, 137.6, 144.4, 144.5, 144.5, 144.6, 147.1, 157.0. HRMS (ESI) calc. for [C_16_H_15_F_3_N_2_S]^+^ [M + H]^+^ 325.0981, found 325.0984.

*2-Methyl-4-(4-nitrophenoxy)-7-(trifluoromethyl)-2,3-dihydro-1H-pyrrolo[3,4-c]pyridine* (**4b**). 73%. M.p. 91–93 °C. ^1^H NMR (300 MHz, CDCl_3_): δ 2.68 (s, 3H, N-CH_3_), 4.09 (s, 2H, N-CH_2_), 4.17 (s, 2H, N-CH_2_), 7.34 (d, *J* = 9.0 Hz, 2H, 4-NO_2_-Ph), 8.28 (s, 1H, C(6)-H), 8.33 (d, *J* = 9.0 Hz, 2H, 4-NO_2_-Ph). ^13^C NMR (75 MHz, CDCl_3_): δ 41.9, 57.0, 59.9, 121.8, 125.2, 125.5, 143.6, 143.7, 143.8, 144.7, 153.0, 158.1, 159.0. HRMS (ESI) calc. for [C_15_H_12_F_3_N_3_O_3_]^+^ [M + H]^+^ 340.0904, found 340.0902.

*Methyl 4-(benzylthio)-2-methyl-2,3-dihydro-1H-pyrrolo[3,4-c]pyridine-7-carboxylate* (**4c**). 64%. M.p. 93–95 °C. ^1^H NMR (300 MHz, CDCl_3_): δ 2.59 (s, 3H, N-CH_3_), 3.83 (s, 2H, N-CH_2_), 3.91 (s, 3H, CO_2_CH_3_), 4.24 (s, 2H, N-CH_2_), 4.56 (s, 2H, S-CH_2_), 7.21–7.42 (m, 5H, Ph), 8.92 (s, 1H, C(6)-H). ^13^C NMR (75 MHz, CDCl_3_): δ 33.6, 42.2, 45.4, 58.2, 61.8, 117.5, 127.3, 128.5, 129.1, 133.8, 137.7, 149.6, 151.0, 157.2, 166.0. HRMS (ESI) calc. for [C_17_H_18_N_2_O_2_S]^+^ [M + H]^+^ 315.1162, found 315.1172.

*Methyl 2-methyl-4-(4-nitrophenoxy)-2,3-dihydro-1H-pyrrolo[3,4-c]pyridine-7-carboxylate* (**4d**). 40%. M.p. 101–103 °C. ^1^H NMR (300 MHz, CDCl_3_): δ 2.66 (s, 3H, N-CH_3_), 3.92 (s, 3H, CO_2_CH_3_), 4.05 (s, 2H, N-CH_2_), 4.33 (s, 2H, N-CH_2_), 7.31 *J* = 8.1 Hz, 2H, 4-NO_2_-Ph), 8.29 (d, *J* = 8.1 Hz, 2H, 4-NO_2_-Ph), 8.65 (s, 1H, C(6)-H). ^13^C NMR (75 MHz, CDCl_3_): δ 29.7, 42.1, 52.2, 57.0, 61.9, 118.4, 121.6, 124.5, 125.5, 144.6, 149.2, 156.9, 158.3, 159.2, 165.1. HRMS (ESI) calc. for [C_16_H_15_N_3_O_5_]^+^ [M + H]^+^ 330.1084, found 330.1079.

*Methyl 2-methyl-4-(pyrrolidin-1-yl)-2,3-dihydro-1H-pyrrolo[3,4-c]pyridine-7-carboxylate* (**4e**). 32%. M.p. 98–100 °C. ^1^H NMR (300 MHz, CDCl_3_): δ 1.94 (t, *J* = 6.6 Hz, 4H, CH_2_CH_2_CH_2_CH_2_), 2.59 (s, 3H, N-CH_3_), 3.65 (t, *J* = 6.6 Hz, 4H, CH_2_CH_2_CH_2_CH_2_), 3.82 (s, 3H, CO_2_CH_3_), 4.15 (d, *J* = 3.6 Hz, 4H, CH_2_-N-CH_2_), 8.62 (s, 1H, C(6)-H). ^13^C ЯMP (75 MHz, CDCl_3_): δ 25.4, 42.3, 48.1, 51.3, 60.6, 61.4, 110.1, 117.8, 150.4, 152.6, 155.5, 166.6. HRMS (ESI) calc. for [C_14_H_19_N_3_O_2_]^+^ [M + H]^+^ 262.1550, found 262.1556.

*Methyl 4-(isobutylthio)-2-methyl-2,3-dihydro-1H-pyrrolo[3,4-c]pyridine-7-carboxylate* (**4f**). 54%. Oil. ^1^H NMR (300 MHz, CDCl_3_): δ 1.40 (d, *J* = 6.6 Hz, 6H, 2CH_3_(i-Bu)), 1.89–2.00 (m, 1H, CH (i-Bu)), 2.61 (s, 3H, N-CH_3_), 3.20 (d, *J* = 6,6 Hz, 2H, CH_2_ (i-Bu)), 3.86 (s, 2H, N-CH_2_), 3.90 (s, 3H, CO_2_CH_3_), 4.24 (s, 2H, N-CH_2_), 8.87 (s, 1H, C(6)-H). ^13^C NMR (75 MHz, CDCl_3_): δ 21.9, 28.6, 28.7, 37.2, 37.7, 37.9, 42.2, 51.6, 51.9, 58.3, 61.8, 112.1, 117.0, 141.7, 149.6, 150.5, 166.0. HRMS (ESI) calc. for [C_14_H_20_N_2_O_2_S]^+^ [M + H]^+^ 281.1318, found 281.1384.

*4-(Benzylthio)-7-chloro-2-methyl-2,3-dihydro-1H-pyrrolo[3,4-c]pyridine* (**4g**). 45%. M.p. 70–72 °C. ^1^H NMR (300 MHz, CDCl_3_): δ 2.59 (s, 3H, N-CH_3_), 3.88 (s, 2H, N-CH_2_), 3.99 (s, 2H, N-CH_2_), 4.49 (s, 2H, S-CH_2_), 7.22–7.41 (m, 5H, Ph), 8.29 (s, 1H, C(6)-H). ^13^C NMR (125 MHz, CDCl_3_): δ 33.9, 42.1, 59.2, 59.9, 123.0, 127.1, 128.4, 129.0, 134.9, 137.9, 146.4, 147.5, 150.2. HRMS (ESI) calc. for [C_15_H_15_ClN_2_S]^+^ [M + H]^+^ 291.0717, found 291.0718.

*7-Chloro-2-methyl-4-(4-nitrophenoxy)-2,3-dihydro-1H-pyrrolo[3,4-c]pyridine* (**4h**). 37%. M.p. 119–121 °C. ^1^H NMR (300 MHz, CDCl_3_): δ 2.65 (s, 3H, N-CH_3_), 4.07 (s, 4H, CH_2_-N-CH_2_), 7.26 (d, *J* = 9.0 Hz, 2H, 4-NO_2_-Ph), 8.00 (s, 1H, C(6)-H), 8.29 (d, *J* = 9.0 Hz, 2H, 4-NO_2_-Ph). ^13^C ЯMP (125 MHz, CDCl_3_): δ 42.0, 58.1, 60.0, 120.7, 122.6, 125.5, 125.9, 144.1, 144.6, 152.8, 155.0, 159.0. HRMS (ESI) calc. for [C_14_H_12_ClN_3_O_3_]^+^ [M + H]^+^ 306.0639, found 306.0639.

*4-(Benzylthio)-7-bromo-2-methyl-2,3-dihydro-1H-pyrrolo[3,4-c]pyridine* (**4i**). 47%. Oil. ^1^H NMR (300 MHz, CDCl_3_): δ 2.63 (s, 3H, N-CH_3_), 4.00 (s, 2H, N-CH_2_), 4.05 (s, 2H, N-CH_2_), 4.41 (s, 2H, S-CH_2_), 7.17–7.32 (m, 5H, Ph), 8.35 (s, 1H, C(6)-H). ^13^C NMR (75 MHz, CDCl_3_): δ 33.9, 42.2, 59.4, 61.5, 111.5, 127.2, 128.5, 129.0, 129.3, 134.8, 137.9, 148.7, 149.4, 151.0. HRMS (ESI) calc. for [C_15_H_15_BrN_2_S]^+^ [M + H]^+^ 335.0206, found 335.0212.

*7-Bromo-2-methyl-4-(4-nitrophenoxy)-2,3-dihydro-1H-pyrrolo[3,4-c]pyridine* (**4j**). 48%. M.p. 108–110 °C. ^1^H NMR (300 MHz, CDCl_3_): δ 2.67 (s, 3H, N-CH_3_), 4.07 (s, 2H, N-CH_2_), 4.14 (s, 2H, N-CH_2_), 7.26 (d, *J* = 9.0 Hz, 2H, 4-NO_2_-Ph), 8.10 (s, 1H, C(6)-H), 8.28 (d, *J* = 9.0 Hz, 2H, 4-NO_2_-Ph). ^13^C NMR (75 MHz, CDCl_3_): δ 42.1, 58.3, 61.7, 110.6, 120.9, 125.6, 144.3, 147.1, 154.5, 155.8, 158.9. HRMS (ESI) calc. for [C_14_H_12_BrN_3_O_3_]^+^ [M + H]^+^ 350.0134, found 350.0128.

*Methyl (3aR,5aR,8aS,8bR)-5-(benzylthio)-2,7-dimethyl-8b-nitro-2,3,3a,6,7,8,8a,8b-octahydrodipyrrolo[3,4-b:3′,4′-d]pyridine-5a(1H)-carboxylate* (**5**). 55%. Oil. Rf = 0.6. ^1^H NMR (300 MHz, CDCl_3_): δ 2.19–2.25 (m, 1H), 2.27 (s, 3H, N-CH_3_), 2.28 (s, 3H, N-CH_3_), 2.51 (d, J = 9.9 Hz, 1H), 2.53 (d, J = 10.0 Hz, 1H), 2.74 (t, J = 8.6 Hz, 1H), 2.99 (d, J = 11.0 Hz, 1H), 3.32–3.39 (m, 2H), 3.58–3.67 (m, 2H), 3.77 (s, 3H, CO_2_CH_3_), 4.12 (d, J = 13.8 Hz, 1H), 4.24 (d, J = 13.8 Hz, 1H), 4.94 (dd, J_1_ = 5.2 Hz, J_2_ = 2.2 Hz, 1H), 7.21–7.35 (m, 5H, Ph). ^13^C NMR (75 MHz, CDCl_3_): δ 34.1, 41.3, 41.5, 43.3, 53.2, 58.5, 58.8, 60.3, 62.5, 62.6, 66.0, 94.2, 127.1, 128.4, 129.0, 129.4, 137.4, 160.2, 170.9. HRMS (ESI) calc. for [C_20_H_26_N_4_O_4_S]^+^ [M + H]^+^ 419.1747, found 419.1747.

## 4. Conclusions

Thus, we showed that 2-substituted 5-*R*-3-nitropyridines and isomeric 3-*R*-5-nitropyridines can act as dipolarophiles in [3+2]-cycloaddition reactions with N-methyl azomethine ylide. The possibility and rate of the process depends on the combination of substituents at positions 2 and 5: the presence of electron-withdrawing groups is required. As a result, a number of new functionally complex derivatives of pyrroline and pyrrolidine condensed with a pyridine ring were synthesized.

## Data Availability

Not applicable.

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
