# Peer review of "Nitropyridines as 2π-Partners in 1,3-Dipolar Cycloadditions with N-Methyl Azomethine Ylide: An Easy Access to Condensed Pyrrolines"

_molecules, 2021, doi:10.3390/molecules26185547_

Round 1

Reviewer 1 Report

I will start from my decision of rejecting this paper because, even if the overall manuscript is well written it is at the same time very confused. there are several part that need to be revised and better explained. It is not clear because in Scheme 2 the pyridine is drawn in some way with R1, R2 and R3 in 2,3 and 4th position, while in the subsequent scheme 3 R1, R2 and R3 are in different positions.

I can also ask to the author whyin Table 1 the entry 2a-q are reported and then in Scheme 3 only 10 final product are reported.

In Scheme 2 B: is reported but the nature of this B is not reported anywhere

Still in Table 1 three different reaction conditions are reported but I can’t understand why! Are there reported in literature or are they new?

Somehow the author reported that with electron donating groups the reaction doesn’t occur but any mechanism and explanation is reported

Finally, concerning the Figure 1, the explanation of the 1H-1H NOESY is not clear. So a further specific analysis should be included.

Regarding the references, It is not clear because in some references the DOI is included and in some others it lacks.

Author Response

On scheme 2 you can see the synthesis of the starting compounds. Since the substituents in them vary in three positions, we used the designations R1, R2, R3. The explanation is given in Table 1. Below the table 1 are the conditions for nucleophilic substitution reactions. These conditions apply to the replacement of the mobile chlorine atom in aromatic compounds.

There is no substituent R3 in Scheme 3. The scheme presents successful examples of the involvement of 2-substituted 5-R-3-nitropyridines in 1,3-dipolar cycloaddition. In text also say about some unfortunate examples of reactions. The reaction with isomeric 3-R-5-nitropyridines is presented in a separate scheme 4. Thus, the number of initial compounds should not coincide with the number of compounds in Scheme 3.

We carried out a detailed study of the cycloaddition adduct using various NMR experiments (COSY, 1H-13C HMBC, 1H-13C HSQC, NOESY). The full assignment of hydrogen and carbon atoms in NMR spectra was made.

Added DOIs for all articles to the list.

Reviewer 2 Report

In this manuscript, the authors have described the effect of substituent on the 1,3-dipolar addition of azomethine ylide to 3,5-disubstiutued pyridines as an extension of their previous work. They did a thorough screening of diverse substrates with substituents at 2, 3, and 5-positions of pyridine. This work portrays a useful strategy to access pharmaceutically relevant fused N-heterocycles (stereoselectively) via [3+2] cycloaddition of azomethine ylide. The research findings presented here are significant. It could be further bolstered by addressing the following aspects.

Comments and Suggestions:

  1. Did the authors observe any steric influence of the 2-substituent in case of the double dipole addition?

  1. Do the authors have ee information for product 5? Did they observe any dearomatization in this case? It would be better to write the reaction conditions in Scheme 3.

  1. Did the authors observe the double cycloaddition product with any other substrates (such as 3-nitro-5-carboxylate)?

  1. Line 26: please correct the “dearomatization” spelling.

Author Response

  1. We did not observe the steric effect of the substituent at position 2 on the double cycloaddition. In the case of the O-C6H4NO2 substituent, we also observed two products. Apparently cis and trans isomers. However, it was not possible to separate them.
  2. Double cycloaddition occurs diastereoselectively.
  3. According to the NMR spectra of the crude reaction mixtures, the products of double cycloaddition were observed in a number of cases in trace amounts.
  4. Line 26 correction is made.

Reviewer 3 Report

In this manuscript, Bastrakow and co-workers described the 1,3-DC preparation of several pyrrolines and pyrrolidines condensed with a pyridine ring. The effect of the substituents on the nitro pyridine dipolarophiles on the cycloaddition process was investigated.

The experimental part appears to be well carried out with all necessary spectroscopic data included.

I recommend the publication after the minor changes noted below have been made:

-Introduction: please add more classical literature about compounds which activity is related to the presence of pyridine rings. See for example the 3-pyridyl on the D ring of steroids a) Jarman, M. et al. J. Med. Chem. 1998,41, 5375. (b) Giacomelli et al. J. Org. Chem. 1999,64, 4985

-Figure 1, compound 5: add hydrogen atom on C-5

-Page seven, lines 204, 222, 239, please check the Russian words

-Please add eluant for each purified compound (4a-j,5) along with the Rf

Author Response

We agree with all the comments. Corresponding changes have been made to the text of the article.